# BRIDGING SUB-TASKS TO LONG-HORIZON TASK IN HIERARCHICAL GOAL-BASED REINFORCEMENT LEARNING

## ABSTRACT

Hierarchical goal-based reinforcement learning (HGRL) is a promising approach to learn a long-horizon task by decomposing it into a series of subtasks of achieving subgoals in a shorter horizon. However, the performance of HGRL crucially depends on the design of intrinsic rewards for these subtasks: as frequently observed in practice, short-sighted reward designs often lead the agent into undesirable states where the final goal is no longer achievable. One potential remedy to the issue is to provide the agent with a means to evaluate the achievability of the final goal upon the completion of the subtask; yet, evaluating this achievability over a long planning horizon is a challenging task by itself. In this work, we propose a subtask reward scheme aimed at bridging the gap between the long-horizon primary goal and short-horizon subtasks by incorporating a look-ahead information towards the next subgoals. We provide an extensive empirical analysis in MuJoCo environments, demonstrating the importance of looking ahead to the subsequent sub-goals and the improvement of the proposed framework applied to the existing HGRL baselines.

## 1 INTRODUCTION

A variety of real-world sequential decision-making problems can be conceptualized as the pursuit of specific goals, e.g., navigating a walking robot (Schaul et al., 2015; Nachum et al., 2018) or moving an object using a robotic arm (Andrychowicz et al., 2017) to a specific position. Goal-conditioned reinforcement learning (RL) addresses these challenges by developing goal-conditioned policies that maximize returns with respect to the target goal. This approach offers a versatile policy applicable to a range of distinct problems, each defined by its respective goal. Besides this, goal-conditioned RL enables hierarchical goal-based RL (HGRL) (Eysenbach et al., 2019; Huang et al., 2019; Zhang et al., 2020; Kim et al., 2021; Lee et al., 2022), decomposing a complex long-horizon goal into a series of manageable short-horizon subgoals.

However, the efficacy of HGRL crucially hinges on the design of rewards associated with these subtasks. Empirical observations highlight that myopic reward designs often misguide the agent, leading it to unfavorable states where the ultimate goal becomes unattainable, despite being good at accomplishing subgoals. We illustrate this problem with Figure 1: in (A), the agent achieved subgoal $sg_1$ and $sg_2$ while it could not process to the final goal $g$ since it tumbled around $sg_2$; but in (B) it achieved all the subgoals and final goal. In both (A) and (B), a myopically designed reward function would give the same reward for the segment between $sg_1$ and $sg_2$. Such a myopic reward often misguides the agent to learn how to tumble around $sg_2$ from (A).

In particular, this problem can be intensified by hindsight experience replay (HER) (Andrychowicz et al., 2017), which is a common practice for further acceleration of HGRL. The key idea of HER it to augment data by transforming any trajectory into a virtually successful one by shifting subgoals or goals. Hence, it would generate more augmented segments with subgoal $sg_2$ using (A) rather than (B) since the agent struggled around $sg_2$ quite a while in (A) but it just passed $sg_2$ in (B). Consequently, this misguides the agent on $sg_2$ whereas HER hindsights the beneficial experience to $sg_1$ in (A).

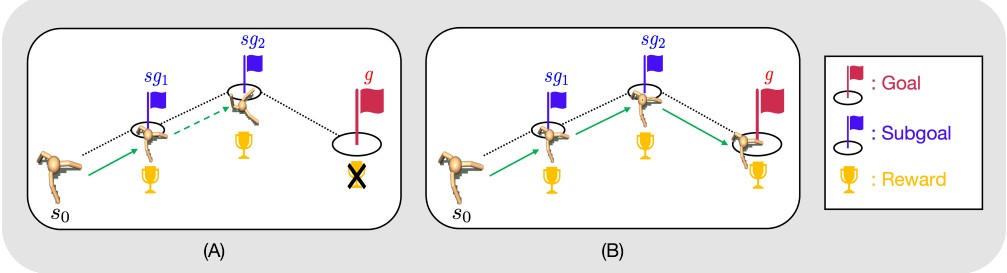

Figure 1: **A motivating example.** In both (A) and (B), a tripod agent achieved $sg_2$, whereas it malfunctioned in the end of (A). Hence, to mitigate adverse impact of learning (A), we need a forward-looking reward that discriminates the segments to $sg_2$ in (A) and (B) by assessing the feasibility of achieving the final goal in advance.

The challenge lies in the intricate balance between providing the necessary guidance to reach subgoals and ensuring that these intermediate accomplishments align with the pursuit of the final goal. To mitigate this issue, providing the agent with a mechanism to estimate the feasibility of reaching the final goal upon completing a subgoal seems promising. However, such an estimation would be unavailable or unstable in the beginning, in particular, when the goal is challenging. Paradoxically, the estimation is more demanding as the ultimate goal is more difficult. In addition, the design of reward depending on both subgoal and goal necessitates the policy being aware of them, which dilutes the benefit from the goal decomposition in HGRL. This motivates us to devise an effective proxy for the feasibility of the ultimate goal.

To this end, we devise a simple yet effective design of reward for subgoals, called forward-looking reward (FLR), which is also easily applicable to most of existing HGRL methods. To be specific, we let the agent simply check if it can proceed to a potential next subgoal slightly distant from the current subgoal, instead of assessing the feasibility of the final goal. By doing so, we guide the agent towards each subgoal while favoring states and actions conducive to accomplishing subsequent subgoals on the path to the final objective.

To substantiate our proposed approach, we present an extensive empirical analysis conducted within the MuJoCo simulation environments. Our results underscore the critical importance of considering and incorporating a forward-looking perspective toward subsequent sub-goals in the sub-task reward design. We also verify the efficacy of the proposed short-term look-ahead as an affordable proxy for the long-term look-ahead. Furthermore, we demonstrate the superior performance of our proposed framework over existing HGRL baselines, in particular, when the training begins with unskilled agents even for subgoals. This affirms the effectiveness of our forward-looking approach in facilitating more efficient HGRL as we enable HGRL with immature agents, while the previous methods (Kim et al., 2023; Lee et al., 2022; Kim et al., 2021) often require a certain level of proficiency.

Our main contributions are summarized as follows:

- We show that the previous HGRL methods overlook the importance of forward-looking rewards for subgoals, and thus often lead to short-sighted agents, just optimizing immediate subgoals while neglecting the ultimate goal in future.

- To address this inefficiency, we introduce FLR provisioning an effective proxy of the feasibility of the final goal. FLR is simply applicable to most of HGRL methods.

- The extensive experiments analyze the problem of not forward-looking in HGRL, and justify the proposed FLR, by demonstrating the improvement of the proposed framework applied to the existing HGRL baselines, in particular, when starting from unskilled agents.

**Related works**   As mentioned, designing subtasks is one of the most prominent components in hierarchical RL, to make subtasks quickly affordable while minimizing the risk of misalignment to the ultimate task. To this end, there have been pioneer works to learn or meta-learn such subtasks (Nachum et al., 2018; Florensa et al., 2017; Vezhnevets et al., 2017; Li et al., 2020) or directly sub-policies (Vezhnevets et al., 2016; Bacon et al., 2017), and to define subtasks directly using the

advantage function comparing the values of the current and previous states to complete the task (Li et al., 2019). Meanwhile, the goal-based RL naturally provides a form of subtasks, described by subgoals sharing the same space of goals, where the goal is defined by a desired feature of states. Despite the misalignment issue illustrated in Figure 1, this is a reasonable practice as the goal would be carefully tailored and specified by users, and it requires no extra efforts to seek other subtasks (Nachum et al., 2018; Florensa et al., 2017; Vezhnevets et al., 2017; Li et al., 2020). Hence, in this work, we aim to provide an effective heuristic to bridge the misalignment gap from inheriting the predefined description of goals as that of subgoals.

## 2 METHOD

### 2.1 PROBLEM FORMULATION

We consider a goal-conditioned Markov decision process (gMDP), defined by a tuple $\mathcal{M} := \langle \mathcal{S}, \mathcal{A}, p; \mathcal{G}, r, \gamma; \rho_0, \rho_g \rangle$, where $\mathcal{S}$ is the state space, $\mathcal{A}$ is the action space, $p(s_{t+1}|s_t, a_t)$ is the (latent) stationary dynamics function, $\mathcal{G}$ is the goal space, $r$ is the (extrinsic) goal-dependent reward function, $\gamma \in (0, 1)$ is the discounted reward, $\rho_0$ is the initial state distribution, and $\rho_g$ is the goal distribution. The goal is often described by a desired feature of states (Kim et al., 2023; Lee et al., 2022; Kim et al., 2021). To be specific, the goal is often encoded in reward function $r$ for $r_{t+1}$ as follows:

$$r(s_{t+1}, g) := \begin{cases} 0 & \text{if} \quad \|\phi(s_{t+1}) - g\| < \delta \\ -1 & \text{otherwise} \end{cases}, \tag{1}$$

where $\phi$ is a feature extractor (e.g., in navigation tasks, $\phi(s_{t+1})$ is the agent location while state $s_{t+1}$ contains more information than the location.) and $\delta > 0$ is a threshold determining the success of achieving a goal.

The reward design given in (1) guides the agent to obtain the desired feature (that matches the goal) and maintain it. We aim to find a policy $\pi : \mathcal{S} \times \mathcal{G} \mapsto [0, 1]^\mathcal{A}$ that maximizes the following objective $J(\pi)$:

$$J(\pi) := \mathbb{E}_{\rho_0, \rho_g} [V_\pi(s_0; g)], \qquad V_\pi(s_0; g) := \mathbb{E}_{\pi, g} \left[ \sum_{t=0}^{\infty} \gamma^t r_{t+1} \mid s_0 \right] \tag{2}$$

where the expectation $\mathbb{E}_{\rho_0, \rho_g}$ is taken over the initial state distribution $\rho_0$ and the goal distribution $\rho_g$, and $V_\pi(s_0; g)$ is the value function of goal-conditioned policy $\pi(\cdot; g)$ for goal $g$.

**Policy hierarchy** A standard hierarchy of policy $\pi = (\pi_h, \pi_l)$ consists of high-level policy $\pi_h : \mathcal{S} \times \mathcal{G} \mapsto \mathcal{G}$ and low-level policy $\pi_l : \mathcal{S} \times \mathcal{G} \mapsto [0, 1]^\mathcal{A}$ on different time horizons. Specifically, at every $k$ steps, the high-level policy $\pi_h(s_{ki}, g)$ chooses $i$-th subgoal $sg_i \in \mathcal{S}$ to guide the low-level policy for the next $k$-steps, i.e., action $a_t \sim \pi_l(s_t, sg_i)$ for $t \in [ki, ki + k)$. This enables a decomposition of the RL problem in (2) into ones over shorter horizons: (i) high-level RL for $\pi_h$ over transitions $\{(s_{ki}, sg_i, r_{i+1}^h, s_{ki+k})\}_{i=0}^{\infty}$ of a high-level horizon indexed by $i$; and (ii) low-level RL for $\pi_l$ on $k$ transitions $\{(s_t, a_t, r_{t+1}^l, s_{t+1})\}_{t=ki}^{ki+k-1}$, where the reward $r^h$ and $r^l$ are the key design components to reduce the discrepancy between the original and decomposed problems. For the ease of presentation, we consider an HGRL framework that repeatedly alternates two different RL problems to improve high-level $\pi_h$ and low-level $\pi_l$, respectively, considering the other is fixed.

The key challenge is to design the *intrinsic* low-level reward $r^l$, and we propose our method for designing $r^l$ in the next section. For the high-level policy, we can define the high-level reward as $r_{i+1}^h := \sum_{t=0}^{k-1} \gamma^t r_{ki+t+1}$ with a discount factor $\gamma^k$. We can easily check the equivalence between the original value function $V_{(\pi_h, \pi_l)}(s; g)$ and the high-level one $V_{\pi_h}^h(s; g, \pi_l) := \mathbb{E}_{(\pi_h, \pi_l), g}[\sum_{i=0}^{\infty} \gamma^{ki} r_{i+1}^h]$, and thus between the original objective $J((\pi_h, \pi_l))$ and the high-level objective function $J_h(\pi_h; \pi_l) = \mathbb{E}_{\rho_0, \rho_g} [V_{\pi_h}^h(s_0; g, \pi_l)]$. Besides using the rewards from the environment for the high-level policy $\pi_h$ as in (Kim et al., 2021; Zhang et al., 2020; Nachum et al., 2018), we can also employ a shortest path algorithm on a graph of goals as $\pi_h$ (Kim et al., 2023; Nasiriany et al., 2019; Huang et al., 2019). In our experiments, we use the latter formulation.

## 2.2 Forward-looking Reward Design for Low-level Policy

In this section, we propose the *forward-looking* reward (FLR) shaping that guides the low-level agent to consider the achievability of the final goal with respect to the current subgoal or the state. The key difference of our reward shaping from the existing ones, which use the form (1) as is, is to exploit the future outcomes *after* reaching the given subgoal, i.e., it is forward-looking. To this end, we first describe an idealized reward function that assumes access to the true value function of the high-level policy. Next, we describe our practical implementation to estimate the value function using a proxy model.

**Reward with true value function.** We first describe the desired reward function when one has access to the true value function, i.e., the expected reward upon reaching the given subgoal. To be specific, we consider reward design for the low-level policy $\pi_l$ on the $k$ transitions $\{(s_t, a_t, r_{t+1}^l, s_{t+1})\}_{t=ki}^{ki+k-1}$ given subgoal $sg_i = \pi_h(s_{ki}, g)$ from the high-level policy for goal $g$, as follows:

$$r_\alpha(s_{t+1}, sg_i, g) := \begin{cases} \alpha V_\pi(s_{t+1}; g) & \text{if} \quad \|\phi(s_{t+1}) - sg_i\| < \delta \\ -1 & \text{otherwise} \end{cases}, \tag{3}$$

where $V_\pi(s_{t+1}; g)$ is the value of state $s_{t+1}$ under policy $\pi = (\pi_h, \pi_l)$ for the final goal $g$. The intention behind this reward design $r_\alpha$ is to drive the low-level policy for subgoal $sg_i$ while preferring states that are advantageous to achieve the final goal $g$.

**The choice of $\alpha$** The choice of $\alpha$ controls the priority on the subgoal over the final goal. Note that the reward $r_\alpha$ with $\alpha = 0$ reduces to the reward in (1), which neglects the final goal. However, as illustrated in Figure 1, the achievability of subsequent subgoals $sg_{i+1}, sg_{i+2}, \ldots$, and the final goal $g$ must be taken into account to successfully complete the original long-horizon task.

Therefore, we set $\alpha > 0$, which guides the low-level agent to the favorable states to the final goal $g$ after reaching the subgoal $sg_i$. However, noting that $V_\pi(s_{t+1}; g) \leq 0$, overly large value of $\alpha$ would result in a misleading reward such that $-1 > \alpha V_\pi(s_{t+1}; g)$, demotivating the agent to achieve the subgoal. To eliminate such a misleading rewarding, we propose a small but positive value of $\alpha$, in particular, bounded by $(1 - \gamma)$. The upper bound $(1 - \gamma)$ is from the lower bound of the value function, $V_\pi(s_{t+1}; g) \geq -\frac{1}{1-\gamma}$. With the choice of $\alpha \in (0, 1-\gamma]$, we can guarantee that the reward is always informative, i.e.,

$$-1 \leq \alpha V_\pi(s; g), \tag{4}$$

which ensures that the reward function keeps attracting the low-level agent around the subgoal curated by the high-level policy. The same intention can be alternatively implemented as $r_{t+1}^l = r_\beta(s_t, s_{t+1}, g) = \gamma V_\pi(s_{t+1}; g) - V_\pi(s_t; g) + \beta r(s_{t+1}, sg_i)$ that continuously consults the value function, c.f., (Li et al., 2019). However, such frequent access to the value function might be prone to unstable learning as the agent simultaneously learns the value function.

**Forward-looking reward (FLR) with a proxy for $V_\pi(s; g)$** Referring the value function $V_\pi(s; g)$ provides a beneficial provision to the low-level policy on the final goal $g$. However, the estimate of $V_\pi(s; g)$ is not informative at the beginning, in particular, when goal $g$ is challenging, c.f., Figure 5(b). In addition, to learn RL problems associated with $r_\alpha(s, sg_i, g)$ in (3), we may need to redefine the low-level policy as $\pi_l(s, sg_i, g)$ rather than $\pi_l(s, sg_i)$ agnostic to $g$. This requires an additional complication to train the low-level policy.

These limitations of $r_\alpha$ with $V_\pi(s; g)$ motivate us to devise a proxy of $V_\pi(s; g)$ without specification on $g$. For an effective proxy, we propose to replace the final goal $g$ with a potential next subgoal in an affordable range. Specifically, the proposed reward for $r_{t+1}^l$ is given as follows: for some $\varepsilon > 2\delta$,

$$\tilde{r}_\alpha(s_{t+1}, sg_i) := \begin{cases} \max_{sg:\varepsilon \leq \|sg_i - sg\|} \alpha V_{\pi_l}^l(s_{t+1}; sg) & \text{if} \quad \|\phi(s_{t+1}) - sg_i\| < \delta \\ -1 & \text{otherwise} \end{cases}, \tag{5}$$

where $V_{\pi_l}^l(s, sg) := \mathbb{E}_{\pi_l, sg}\left[\sum_{t=0}^k \gamma^t \tilde{r}_\alpha(s_{t+1}, sg) | s_0 = s\right]$ is the value function of policy $\pi_l(\cdot, sg)$ initialized at state $s$ for subgoal $sg$. The maximization over subgoals in (5) yields the subgoal

$sg^* = \arg\max_{sg:\varepsilon \leq \|sg_i - sg\|} V^l_{\pi_l}(s_{t+1}, sg)$ that is the most confident among subgoals distant from the current $sg_i$. The choice of $\varepsilon > 2\delta$ ensures that $sg^*$ is not achieved at $s_{t+1}$ as

$$\|sg^* - \phi(s_{t+1})\| \geq \|sg^* - sg_i\| - \|sg_i - \phi(s_{t+1})\| \geq \varepsilon - \delta > \delta . \tag{6}$$

The selection of $sg^*$ with the most confidence can be interpreted as a plausible next subgoal since in HGRL, the high-level policy would plan a path of subgoals to the final goal using the confident transitions from one to another. Presuming that more distant goal is more difficult Zhang et al. (2020); Kim et al. (2021), being achievable to subgoal $sg^*$ slightly different than the current $sg$ is a necessary criterion to being achievable to final goal $g$ likely further than the current $sg_i$. In addition, the estimation of $V^l_{\pi_l}(s_{t+1}; sg^*)$ for the most confident next subgoal $sg^*$ would be more stable than $V_\pi(s_{t+1}; g)$ for the final goal $g$. Hence, the estimate of $V^l_{\pi_l}(s_{t+1}, sg^*)$ is a good representative of the reachability to nearby subgoals. The reward $\tilde{r}_\alpha$ can guide the low-level agent to the current $sg_i$ while being reachable to the neighboring subgoals. It is also worth to mention that the same upper bound $(1 - \gamma)$ of $\alpha$ can guarantee no misleading reward in (4).

## 2.3 HGRL with Forward-Looking Reward

Drawing from FLW, we introduce a practical training algorithm for the low-level policy that seamlessly integrates with the latest HGRL methodologies. (Lee et al., 2022; Kim et al., 2021; 2023). Moreover, within the context of HGRL, we incorporate the use of hindsight experience replay (HER) Andrychowicz et al. (2017) to address the challenges posed by sparse reward settings.

In this process, we begin by sampling a batch of transitions $\{(s_t, a_t, s_{t+1}, sg_t)\}_{|B|}$ from the replay buffer, where $|B|$ represents the batch size. Subsequently, we leverage the principles of HER to re-label the sub-goals, denoted as $sg_t$, by selecting one of the successfully achieved states $\phi(s_{t+f})$ within the trajectory, where $f$ is the sampled future step from current timestep $t$. Following the re-labeling step, we proceed to compute updated rewards using (5). This involves (i) identifying transitions within the batch that effectively reach the re-labeled sub-goals, (ii) sampling several next sub-goals, $sg'$, at a distance of $\varepsilon$ from the original sub-goals for these selected transitions and finding $sg'$ that maximizes the value as mentioned in (5), and (iii) assigning a reward of $V^l_{\pi_l}(s_{t+1}; sg')$ for these transitions, while assigning a reward of -1 for others. Finally, we proceed to update the low-level policy accordingly. For a detailed pseudo-code of our algorithm, please refer to Algorithm 1 in Appendix A.

# 3 Experiments

We demonstrate the proposed forward-looking reward design in the Mujoco environments (Todorov et al., 2012). We note that FLR can serve as an extension to all HGRL techniques without look-ahead mechanisms. For a plug-in method, we adopt the DHRL method proposed in Lee et al. (2022) which we describe in the next subsection in detail. Hence, we refer to our proposed method as (FLR+DHRL) in all our experiments.

## 3.1 Experiments Setup

**Environments**  We primarily conduct our experiment on the Ant Maze navigation task introduced in Todorov et al. (2012). In this environment, an ant must navigate to various locations in a large sized maze. We consider two types of environments and agents:

- AntMaze4leg: This is the standard Ant-Maze environment with 4-legged ant starting at the bottom left in $24 \times 24$, $\supset$-shaped maze, and targets to the top left of the corner.
- AntMaze3leg: 3-legged ant starts at the left in $24 \times 8$ room, and targets to the right. This setup adds complexity compared to traditional locomotion tasks and tests the adaptability and robustness of our approach.

**Baselines**  We compare our method (FLR+DHRL) to the previous HGRL methods described as the following:

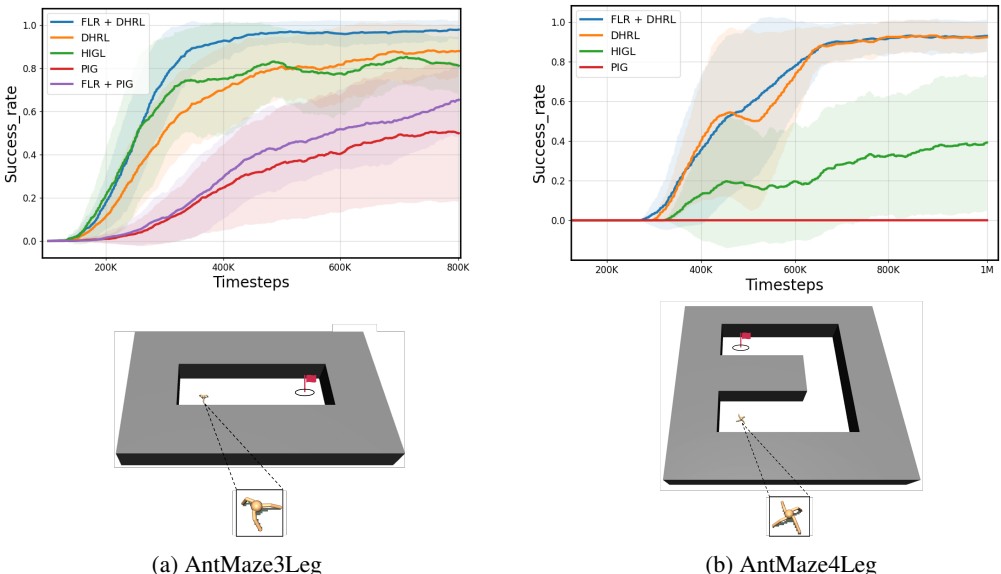

(a) AntMaze3Leg          (b) AntMaze4Leg

Figure 2: Investigating success rate across AntMaze3Leg and AntMaze4Leg: We use 5 random seeds, unveiling average performance and standard deviation smoothed equally. Note that the success rate of PIG surges after 1M timesteps in AntMaze4Leg.

- HIGL (Kim et al., 2021): HIGL is a method in which the high-level policy is guided by landmarks obtained through graph planning with $k$-step constraint. The low-level policy is trained with dense rewards.(i.e. $r_l(s_{t+1}, g) = -\|\phi(s_{t+1}) - g\|_2$)

- DHRL (Lee et al., 2022): DHRL is a method in which the sequence of waypoints is made by the subgoal generated by a high-level policy and graph planning algorithm. The low-level is trained to follow the waypoint of the shortest path with sparse rewards using HER.

- PIG (Kim et al., 2023): PIG is a method in which the target-goal conditioned policy imitates the sub-goal conditioned policy, where the sub-goal is generated by the graph planning. Sparse reward with HER is used for training target policy.

**Our method** As previously mentioned, our proposed FLR scheme is broadly applicable to all HGRL methods. In our experiment, we attach FLR to DHRL and PIG in AntMaze3Leg to show the improvement and applicability of FLR to HGRL and analyze with DHRL due to the outstanding performance exhibited by DHRL in recent HGRL benchmarks (for the detailed exposition of DHRL, we refer the readers to Algorithm 2 in Appendix A). For all environments, we use the reward threshold $\delta = 0.5$, $\alpha = 0.01$, and the sampling distance $\varepsilon = 2$. More detailed information on our experiments (*e.g.,* hyper-parameters, rollouts, etc.) can be found in Applendix B.

**Evaluation** We evaluate the success rate, which signifies the number of times the agent succeeded over the course of 10 test episodes. Similar to the evaluation criteria applied by previous baselines, we define the threshold of 'success' as when the agent enters within a radius of 5 from the final goal in both experimental experiments. We also cumulatively assess the number of episodes in which the agent ends in a fallen state during the testing process.

## 3.2 PERFORMANCE AND EFFICIENCY ANALYSIS

**FLR is more sample efficient** Figure 2 shows the result of our method compared with others. As we observe, FLR demonstrates superior performance in AntMaze3Leg, where a more delicate reward design is necessary to prevent the agent to fall after achieving a sub-goal. Even in AntMaze4Leg, where the chances of falling are relatively lower, our framework exhibits slight gains in performance.

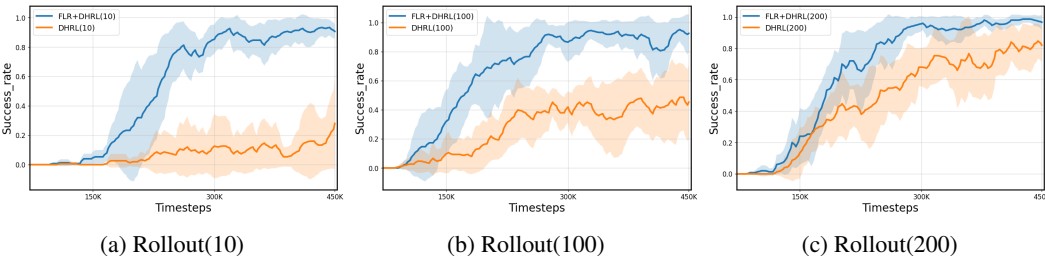

(a) Rollout(10)  (b) Rollout(100)  (c) Rollout(200)

Figure 3: Robustness to the number of initial random rollouts in AntMaze3Leg

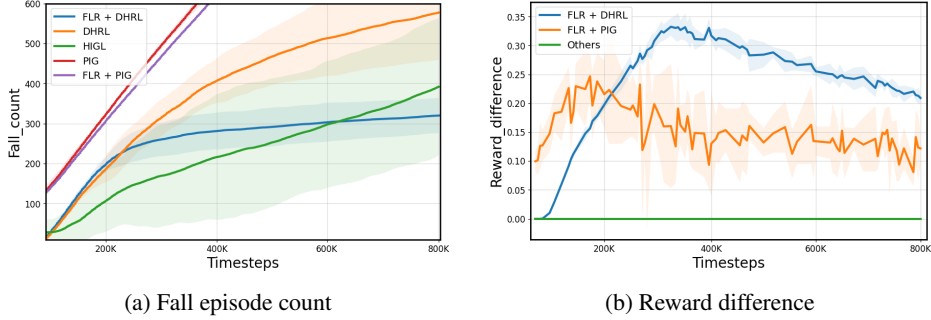

(a) Fall episode count  (b) Reward difference

Figure 4: Validation of FLR in AntMaze3Leg: (a) Our evaluation involves tracking the $z$-positions of ant robots at the last timestep of the episode, specifically counting where it falls below a certain threshold. (b) We computed the reward difference by averaging rewards for transitions reaching sub-goals after applying FLR. Notably, the previous reward function always gets the same reward agnostic to its state.

**FLR-extension reinforces the baseline HGRL method**   The $\supset$-shape in the AntMaze4Leg environment is a challenging obstacle to overcome; only DHRL can effectively accomplish the final goal, as shown in Figure 2. Our method (FLR+DHRL) inherits this ability of DHRL, and outperforms all baselines in both environments. Furthermore, in the AntMaze3Leg environment, we compare performances of the PIG and the FLR-guided PIG (FLR+PIG). As we observe in Figure 2, FLR improves the performance of the vanilla PIG, supporting our claim that FLR can potentially reinforce a broader class of HGRL methods. This superior performance can be attributed to our low-level policy being trained to look ahead subsequent subgoals.

**FLR intrinsically encourages exploration**   As shown in Figure 3, our framework enhances the sample efficiency via giving more intrinsic rewards from the reward look-ahead design. We emphasize that a vanilla DHRL requires a much longer burn-in period through initial random rollout. In contrast, our framework demonstrates robust learning, even when starting with less exploratory trajectories, highlighting its sample-efficient nature.

**FLR leads the agent to stable states**   We report the number of the episodes where the agent falls in Figure 4(a). It is evident that both PIG and DHRL experienced a substantial number of episodes where the agent falls, and FLR can always improve the situation. This indicates that our approach encourages the agent to consider future tasks in the learning process. Moreover, Figure 4(b) illustrates the look-ahead reward difference between the normal and fall states when the proposed FLR is applied. This result demonstrates the effectiveness of FLR in gradually penalizing fallen states as learning progresses, meaning that the agent is trained to look ahead.

## 3.3  ABLATION STUDY

We ablate two components in the proposed FLR scheme: methods to select the next sub-goal and using different values of $\alpha$.

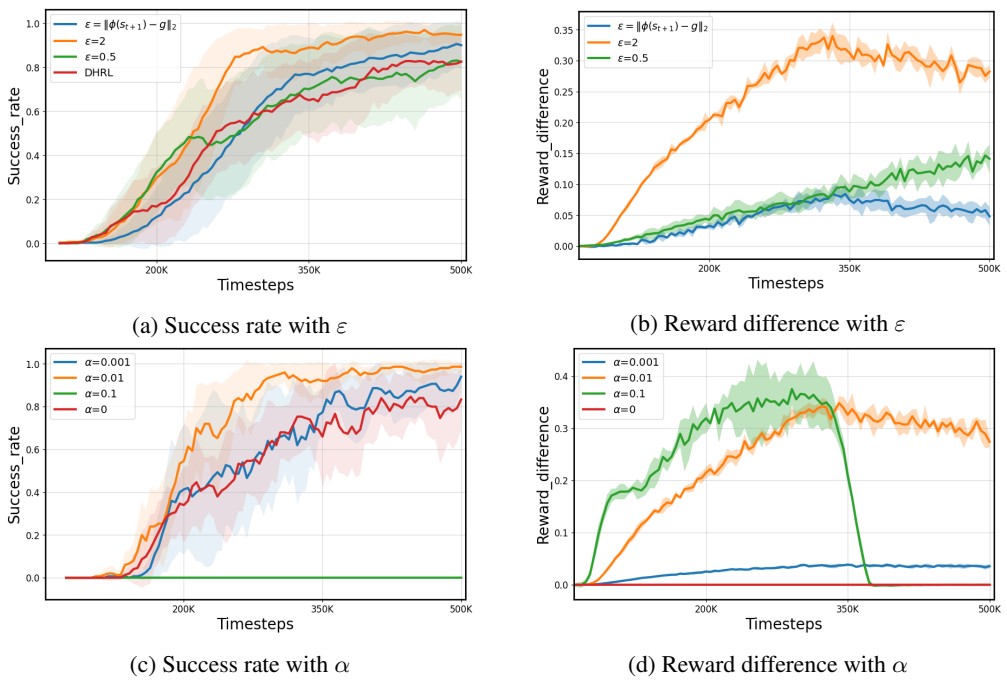

Figure 5: Ablation studies on FLR

**Selecting next sub-goal** We assess several options to select the next sub-goal within our approach: (i) selecting the final goal as the next sub-goal and (ii) setting sampling distance $\varepsilon = 0.5$ which is the same as the reward threshold $\delta$. Figure 5(a) shows that any method of selecting the next sub-goal that is more than $2\delta$ away from the current subgoal outperforms DHRL. Conversely, the method selecting closed subgoals demonstrates comparable or marginally inferior performance. Comparing the performance of selecting next subgoal as goal and $\varepsilon = 2$, we observe that selecting the final goal as the next sub-goal results in relatively lower performance. This is because when the final goal is chosen as the next sub-goal, the value function is only updated after reaching the goal, which leads to difficulty distinguishing between unstable and stable states. As shown in 5(b), selecting too far or near the next subgoal does not make the low-level policy distinguish the fallen states or not. Therefore, in increasingly complex environments, the method of selecting the next sub-goal becomes even more crucial, a topic we leave for future work.

**Effective of $\alpha$** We investigate the role of $\alpha$ in our method. In line with the investigation of the optimal value of alpha mentioned in Section 2.2, selecting the appropriate $\alpha$ is crucial. To investigate its effects, we set $\gamma = 0.99$ and use different values of $\alpha$ in [0.001, 0.01, and 0.1]. Figure 5(c) illustrates that setting alpha to $0.01 = 1 - \gamma$ results in the best performance. This is expected in our reward design: when alpha is too large, even after a certain degree of learning, reaching sub-goals results in receiving rewards less than -1, leading to deteriorated performance. Conversely, when alpha is excessively small, the low-level policy struggles to distinguish between states where the agent has toppled over and those where it has not, resulting in sub-optimal performance.

## 4 CONCLUSION

In summary, our contributions include the recognition of the importance of forward-looking rewards in HGRL, the introduction of forward-looking reward (FLR) as a practical and adaptable solution, and empirical evidence showcasing the superiority of our framework over existing HGRL baselines. Recalling that the previous HGRL requires substantial pre-training of agents for a certain level of proficiency in advance, we believe that the proposed FLR is a simple yet effective practice for more efficient HGRL as it enables HGRL with virtually zero-skilled agents.

### ACKNOWLEDGMENTS

Use unnumbered third-level headings for the acknowledgments. All acknowledgments, including those to funding agencies, go at the end of the paper.

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
