# OpenReview forum: "Bridging Sub-Tasks to Long-Horizon Task in Hierarchical Goal-Based Reinforcement Learning"
_ICLR.cc/2024/Conference — ICLR 2024 Conference Withdrawn Submission_

### Official Review · Reviewer_xgwB · 2023-10-29

**Soundness:** 2 fair
**Presentation:** 2 fair
**Contribution:** 2 fair
**Rating:** 3
**Confidence:** 3

**Summary:**

This paper proposes a novel intrinsic reward function design method in hierarchical goal-based reinforcement learning. In order to alleviate the short-sighted reward designs that often lead the agent into undesirable states where the final goal is no longer achievable, the forward-looking reward (FLR) proposed in this paper can enable the agent to evaluate the achievability of the next subgoal upon the completion of the subtask. FLR can be applied to existing hierarchical goal-based reinforcement learning methods, and the results of the corresponding variants obtained in the Mujoco environment show that FLR can improve the performance of the vanilla algorithm.

**Strengths:**

1. The paper is easy to follow. The description of motivation is clear and makes people feel reasonable and meaningful. It is indeed a problem worthy of attention that the intrinsic reward function designed in hierarchical goal-based reinforcement learning is too short-sighted.
2. The chosen experimental environment is reasonable. If the reward function designed for the agent in the AntMaze3Leg scenario is too short-sighted, it will indeed lead to task failure.
3. The analysis of the experimental results is sufficient. In the paper, the author gives the number of episodes where the agent falls and the look-ahead reward difference between the normal and fall states when the proposed FLR is applied, which clarify the role of FLR in reinforcement learning to a certain extent.

**Weaknesses:**

1. The difference between the performance of the algorithm using FLR and the vanilla algorithm is not significant. Especially in AntMaze4Leg, the performance of FLR+DHRL is almost the same as that of the original DHRL.
2. The experiment was only conducted in two scenarios. I think this is a more empirical paper and should be experimented in more different domains. The current experimental results are not convincing enough.
3. I think FLR is more about improving the original algorithm capabilities, rather than really giving it the ability to solve short-sighted problems. For example, in AntMaze3Leg, as training progresses, PIG cannot effectively suppress the number of episodes where the agent falls, and neither can FLR+PIG.
4. The code is not available, which is not conducive to the reproduction of experimental results.

**Questions:**

1. Why do the curves in Figure 4(b) show a downward trend in the later period?
2. How does FLR+PIG perform in AntMaze4Leg?
3. I think the comparison between FLR+DHRL with oracle-based DHRL in Appendix Figure 6 cannot be directly used to illustrate that FLR encourages the avoidance of undesirable transitions. Is the reason the author gives this conclusion because their final winning rates are similar? How about the learning curve for the original DHRL?

---

### Official Review · Reviewer_oQAw · 2023-10-30

**Soundness:** 2 fair
**Presentation:** 1 poor
**Contribution:** 2 fair
**Rating:** 5
**Confidence:** 4

**Summary:**

This paper investigates the issue of inefficient exploration in hierarchical goal-conditioned reinforcement learning (HGRL). The main idea proposed in this paper is to use a forward-looking reward (FLR) that guides the low-level policy to consider the achievability of the final goal with respect to the current subgoal or state. The FLR is based on a proxy model that estimates the value function of the next subgoal slightly distant from the current one. The paper claims that FLR can improve the performance and sample efficiency of HGRL, especially when the agent starts from unskilled states. The paper also claims that FLR can be easily applied to existing HGRL methods as an extension. Moreover, the empirical analysis on MuJoCo environments shows that FLR outperforms the selected baselines and demonstrates robustness and stability in complex tasks.

**Strengths:**

- It tries to address an important and challenging problem of reward design in HGRL, which has not been well studied in the literature.
- It proposes a simple yet effective solution incorporating a forward-looking perspective and a proxy model for value estimation.

**Weaknesses:**

- The authors claim that their proposed method can work as a plug-in for existing HGRL methods. However, the experiments are conducted only on top of DHRL. Since RL algorithms often perform high-performance variance in different environments/tasks, I think it is necessary to make comparisons with more baselines. And also, I think it would be better if the authors make comparisons on different environments, not only the Maze.
- Key designs lack of explanation, e.g., how to determine the $sg$ in equation 5? Parameter searching via enumeration, or gradient-based?
- Missing introduction of environment details
- The investigation on the effectiveness of $\alpha$ is not thoroughly, as the authors only test a range of $\alpha$ when $\gamma = 0.99$. More trails should be included.
- The writing is not good enough. For instance, the supplementary should be improved, e.g., there should include some natural language description for the given algorithm tables, and it seems the authors didn't conduct repeated experiments in Figure 6; The paper mentioned FPS algorithm, but what is FPS algorithm? No reference and no introduction; Figure 3 shows the results of "initial random rollouts" settings, but there is no description for this setting.

And I suggest the authors improving the writing further, especially the details of the involved environments. As you should not suppose your readers are all familiar with the related work of goal-conditioned RL.

**Questions:**

1. What is FLW in the begining of section 2.3?
2. What is the upperbound of $\epsilon$?
3. Section 2.3, the authors include a reference to HER in wrong format. It should be a parenthetical citation.
4. Section 2.2, the authors claim "we may need to redeﬁne the low-level policy as $\pi_l(s, sg_i, g)$ rather than $\pi_l(s, sg_i)$ agnostic to $g$. This requires anadditional complication to train the low-level policy". How to do that? I cannot relate the corresponding context in this paper. Could you further explain it?
5. What is the mathematical format of $g$ in your work?

---

### Official Review · Reviewer_mzz8 · 2023-11-01

**Soundness:** 3 good
**Presentation:** 3 good
**Contribution:** 3 good
**Rating:** 3
**Confidence:** 5

**Summary:**

In this paper, the authors focus on the problem that short-sighted reward designs often lead the agent into undesirable states in HGRL. That is, with the completion of a simple given subtask, the agent may be misguided, leading to the ultimate goal becoming unattainable. The authors propose a subtask reward scheme to bridge the gap between the long-horizon primary goal and short-horizon subtasks. The motivation of this paper is reasonable, and there are relatively detailed method analysis.

**Strengths:**

+ The motivation of this paper is reasonable, and the introduction is well-organized.
+ The innovation of this paper is relatively strong.
+ The effectiveness of the proposed method is verified by experiments.
+ The pictures presented in this paper are clear and standardized.

**Weaknesses:**

+ Only in the simplest scenarios, the proposed method has obvious performance improvement. More experiments should be conducted in various scenarios(such as FetchPush, FetchPickAndPlace, AntMazeBottleneck, and UR3Obstacle) to highlight the superiority of the proposed method.
+ As can be seen from the ablation study, the experiment is sensitive to the hyperparameter $\varepsilon$, even in a simple task, not a robust hyperparameter is provided.
+ The proposed solution is only suitable for controlling the local reaching task of the robot, because when achieved goal is subsituted with the location of the object being operated, it has the same hindsight goal for all the goal relabeling using HER, and FLR will not yield results as expected.
+ It seems that the subscripts of $sg_{i}$ and $k_{i}$ are not standardized.
+ At the beginning, the value of $V_{\pi}$ is definitely biased, and these biased values will continue to exist in the experience buffer when updating, which will continue to degrade theperformance, and may be a reason for the unsatisfactory performance on complex tasks.
+ The proposed solution is only suitable for controlling the local reaching task of the robot, because when the operated object is not changed in position，it has the same hindsight goal for all the goal relabeling using HER, and FLR will not yield results as expected.

**Questions:**

See weaknesses.